# Technological Aspects of Variation in Process Characteristics and Tool Condition in Grinding Process Diagnostics

**DOI:** 10.3390/ma16041493

**Published:** 2023-02-10

**Authors:** Wojciech Kacalak, Dariusz Lipiński, Filip Szafraniec, Michał Wieczorowski, Paweł Twardowski

**Affiliations:** 1Faculty of Mechanical Engineering, Koszalin University of Technology, Racławicka 15, 75-620 Koszalin, Poland; 2Faculty of Mechanical Engineering, Institute of Applied Mechanics, Poznan University of Technology, 3 Piotrowo St., 60-965 Poznań, Poland

**Keywords:** grinding process, grinding wheel, tool condition, diagnostics, wear, abrasive grain

## Abstract

The article presents the technological aspects of the diagnostics of grinding processes. The main features of the grinding process and their importance in diagnostic issues are discussed. Selected issues of research and assessment of the condition of the active surface of grinding wheels are presented. The authors pointed out that the parameters used to assess the topography of the ground surfaces do not have sufficient possibilities to differentiate the surface condition of the grinding wheels. This publication draws attention to the possibility of using new dedicated parameters to assess the properties of the grinding wheel surface. These parameters have a high ability to differentiate changes occurring as a result of the abrasion of grain vertices, their chipping or loading of the grinding wheel surface. The methodology of assessing the processes of abrasive grain wear and changes in the shape and dimensions of the grinding wheel, taking into account the probabilistic features of the grinding process, was formulated. The directions for the development of abrasive tools are presented, pointing to hybrid tools with a multi-phase structure, modified by additions of abrasive aggregates. A new research direction has also been formulated on the use of additive technology to produce specialised abrasive tools, including those with built-in process sensors.

## 1. Introduction: Selected Features of Grinding Processes Useful for Use in Diagnostic Tasks

In micro-cutting processes, especially in high-precision machining, many phenomena and factors become important in determining the results of the process. The analysis of phenomena and factors influencing the quality of the grinding process allows us to distinguish the following groups of inaccuracy causes:causes of inaccuracy depending on the grinding parameters—resulting directly from the parameters of the grinding process, affecting the thermal and mechanical deformations of the machining system and its vibrations; by changing the machining parameters, we can influence the effects of these causes of inaccuracy;causes of inaccuracy independent of the grinding parameters—resulting from the characteristic features of the machining unit, the workpiece and the grinding wheel, and therefore not subject to control;causes of inaccuracies changing during grinding—resulting from changes in the macro- and microgeometry of the active surface of the grinding wheel, occurring over time of grinding—controllable to a limited extent;disturbances—immeasurable and uncontrollable quantities; it is possible to reduce the unfavourable impact of disturbances on the grinding process by building systems that compensate for their effects.

The multitude of factors and phenomena occurring during the grinding process makes the development of comprehensive diagnostic systems a task that requires taking into account many factors with a complex mechanism of cumulating effects of their interactions. In this work, the characteristic features of grinding processes, influencing the resultant values of the process and useful in its diagnostics, are indicated (Figure 1).

Significant factors influencing the results of the material removal process by abrasive grains include, among others: the random size and shape of abrasive grains, their random distribution on the tool surface, complex wear processes of the tips and whole abrasive grains. To this should be added the discontinuity of the process of micrograin formation (at the micro- and sub-microscale). Local thermal and mechanical deformations of tools and workpiece material, especially linear and angular displacements of abrasive grains under the influence of cutting resistance, are also important. In abrasive micromachining, as well as in various varieties of precision grinding, the penetration of the cutting edge into the workpiece material is much smaller than the radii of the roundness of its corners and is comparable to the height of the surface roughness in the machining zone. The variability of the penetration of the abrasive grains into the workpiece material is an unfavourable but unavoidable feature of micro-cutting processes. At the same time, it should be noted that the specific energy of machining depends not only on the average values of the parameters of the layers cut by individual edges, but also on the range of variation and distribution of the values of these parameters. In all machining methods, abrasive grains in the contact zone with the workpiece, move tangentially to the machined surface, and their penetration into the material is variable along the cutting path. The variability of the cavity is the result of many factors. Among the most important are:variability of the nominal depth of cut, which depends on the kinematic characteristics of the method [1];the irregularities of the workpiece surface in the machining zone and formation of the side pile-ups in the abrasive grain—workpiece contact zone [2,3];local susceptibility of the workpiece material and abrasive grains [1,4];local variations of the grinding wheel active surface [5,6];vibration of the tool and abrasive grains [7];significant local variation (in the grain interaction zone) of temperature rise [8,9,10,11]; especially when machining at very high speeds, materials with low thermal conductivity produce variation in the properties of the workpiece material in micro volumes compared with the volumes of the cut layers [12,13];macro- and micro-discontinuity of chip and pile-ups formation [14];micro chipping of grains and changes in the grinding wheel active surface [15,16,17].

In front of the abrasive grain, between the bottom of the rake surface and the shear surface of the workpiece material, a zone of material stasis is formed, which cannot be stable under the conditions of very high temperatures and the discontinuity of the alternating process of accumulation of local deformation and separation of the material. This was also confirmed by the results of process modelling, illustrated in Figure 2. There is also no justification for the view, sometimes encountered, that there is a certain stable point in this zone, separating the zones of material flow in the directions above the machined surface, below the grain and in the lateral directions. The randomness of the micro-cutting process is higher the smaller the average cross-sections of the layers cut by individual edges. In grinding processes, there is usually a very complex mechanism of cumulative effects of random, harmonic and monotonic causes of inaccuracy. The deterministic component is higher the greater the effects of thermal and mechanical deformation of the workpiece and the greater the effects of machine tool inaccuracy.

The random component depends most heavily on the wear of the grinding wheel and random causes of kinematic inaccuracy of the technological system, with the distribution of the random component approximating a log-normal distribution. In process modelling and simulation, when a random number generator is needed, one of the model distributions is usually chosen, which can cause errors when data from real processes indicate an unusual form, such as multimodal. The probability of contact between the grains on the surface of an abrasive tool and the material of a workpiece with a specific surface topography depends on the statistical characteristics of the distributions of grain vertices and ordinates of the machined surface [18], the thickness of the layer to be removed, the kinematic characteristics of the process and the susceptibility of the tool (Figure 2). The probability of contact between the edges and the workpiece decreases as successive zones of the grinding wheel surface move over the surface already machined by the zones preceding them. The smaller the thickness of the layer being ground, the lower the probability of contact between the grains and the workpiece. In the case of abrasive tools with higher susceptibility (such as abrasive films and grinding wheels with susceptible bonds), the recesses of the grains are smaller, and the probability of contact decreases to a lesser extent with the passage of machining time. The modelled abrasive machining processes are not stationary processes. Errors resulting from the assumption of stationarity of the processes depend on the starting point and the period of data collection. The aggregate distribution differs from the instantaneous distribution the more the longer the total collection period is and the less stable the systematic components are. These components can cause an increase in the variance of the summary distribution, a change in the asymmetry coefficient and a change in the flattening coefficient. The instability of the random components, on the other hand, causes a change in the form of the distribution.

A characteristic phenomenon in the initial zone of contact between the grain and the workpiece (in a kinematic system with an increasing cavity along the path of the edge) is the discontinuity of the process of starting and stopping micro-cutting, noticeable especially for larger susceptibilities of blade attachment. This is understandable, since even the cutting process with larger cavities, considered a continuous process, is characterised by micro-discontinuities in chip formation (Figure 3 and Figure 4). 

From the microscopic examination, it is clear that the lateral material pile-ups overtake the edge of the blade and are then partially removed, as the surface of the pile-up on the side of the micrograin is smoothly shaped by the grain. The top edge of the pile-ups is sharp and has rounding radii of less than 0.1 m in many places. In micro-cutting processes, a reduction in the coefficient of friction between the grain and the workpiece material makes it more difficult to form a chip, while increasing the coefficient of friction causes favourable changes in the actual angle of attack and a marked reduction in the effective radius of the rounded cutting edge—which increases the possibility of effective cutting with grains that are shallowly embedded in the workpiece material.

Increasing the coefficient of friction between the grain and the workpiece material expands the dead zone. Cutting under such conditions is easier and more similar to chip cutting, as the specific energy decreases resulting in lower temperature increments, which is conducive to reducing tensile stress values in the surface layer, although the micro-cutting process may be less stable. Cutting with a wide edge parallel to the machined surface and located perpendicular to the direction of the main movement, extends the path of lateral material displacement, which makes it more difficult to form pile-ups and favourably reduces the ratio of the volume of lateral ridges to the volume of material removed [3,19]. A concave, or flat, grain face is preferable to a convex one in terms of its ability to separate material. Most clearly, these factors become apparent during micro-cutting with monocrystalline silicon carbide grains, which not only have small corner radii, but also have a flat contact surface and form wide microcracks. Cutting with an edge parallel to the work surface and approximately perpendicular to the direction of the edge movement is advantageous, especially when machining materials with high plasticity. It is then possible to reduce the ratio of the volume of the lateral ridges to the volume of the material being removed. In experimental studies of abrasive machining processes, a high-speed image acquisition station (Figure 5) can be useful, when equipped with a camera that allows recording up to several hundred thousand frames per second with additional high-brightness spot lighting. 

Dedicated optical systems [20,21,22], multi-sensor systems [23] and systems using acoustic signal analysis [24,25] also demonstrate high diagnostic utility. Dedicated algorithms are used to identify process features, including statistical [5] and predictive features and those using artificial neural networks [17,26]. Decision support is provided using advanced modelling and simulation methods [27,28,29,30,31,32,33]. 

The topography of the active surface of an abrasive wheel with certain characteristics affects not only the geometric structure of the ground surface and the intensity of grain chipping, especially at the initial stage of the life, but also the specific energy of machining (Figure 6). A high value of the skewness coefficient of the cross-sectional distribution of the machined layers occurs when the radii of the action of the tops of the active grains vary significantly. This occurs as a result of coarse shaping of the active surface of the tool. As a result of machining, the height of asperities of the machined surface is then excessive, but the unit energy of machining is reduced. The feed speed during grinding with coolant feed has a significant effect on the process energy and temperature field (Figure 7).

## 2. Methodology for Surveying and Assessing the Condition of the Active Surface of Tools

The assessment of the machining potential of grinding wheels results both from the physical properties of the tool (abrasive grains, bond, grinding wheel structure) and from the geometrical parameters of the grinding wheel active surface that change during machining. The mechanical and chemical properties of the grinding wheel do not change much during grinding, while the condition of the wheel’s active surface changes due to wear processes. Three main wear processes can occur simultaneously: abrasion of the grain tips, microchipping and chipping of the grains and loading of the active surface with machining products. Abrasive grain tip grinding contributes to increased forces and specific energy; grain tip chipping is a process of renewing cutting properties, and whole-grain chipping results in deviations in tool dimensions and shape.

Moreover, in the machining of light metal alloys, fragmentary loading of the active surface of grinding wheels with machining products may occur. In formulating an effective methodology for evaluating the topography of the grinding wheel active surface, in terms of detecting adverse changes, the following problems can be distinguished:identifying forms of wear, taking into account that the methodology and measures for evaluating the effects of each form of wear will be different;determining the methods for the analysis of the grinding wheel active surface (advantageously without disassembly the grinding wheel from the grinding machine spindle) using scanning methods (Figure 8) used in micro- and meso-scale analysis [34], light scattering analysis methods [20] or the replica method [4];selection of areas to be measured, and consideration of the variation of micro-cutting conditions and grain wear characteristics in different zones of the active surface (Figure 9);determination of tool life criteria.

The bases for selecting parameters for evaluating the condition of grinding wheel surface topography are different from the bases for evaluating the geometric structure of machined surfaces. In the analysis of grinding wheels, it is important to isolate individual abrasive grains and determine their geometric parameters (Figure 9, Figure 10 and Figure 11) and to identify and monitor changes on the surface associated with individual forms of grinding wheel wear. 

The functional characteristics of the ground surfaces and the grinding wheel surface are significantly different. As a consequence, most of the parameters useful in assessing the surface topography are not useful for assessing the active surface of grinding wheels.

For various tools and machining processes, it is reasonable to adopt a dedicated set of parameters to evaluate the stereometric features of the surface. The parameters included in the assessment set should be a complementary set, ensuring high efficiency of classification, ease of interpretation and technological usefulness. 

The analysis of the shape and distribution of abrasive grain tips and their substitutions occurring during the grinding process is, in many analyses, the basis for process evaluation and prediction of machining results. Data acquisition, describing the active surface of the grinding wheel, is usually accompanied by difficulties due to the low reflectivity of the active surface of grinding wheels and its complex structure. 

Measuring the surface topography of abrasive tools is possible, for example, using a laser scanning confocal microscope LEXT OLS 4000 (Olympus Co., Tokyo, Japan), but the limitation is the maximum height of the object being measured. It becomes necessary to use replicas of the surfaces under study [35] with a previously conducted assessment of the accuracy of the surface reproduction. Specialised materials and instrumentation providing a mapping resolution of 0.1 µm can be used to make replicas of the active surface. In order to extract fragments of the corners of abrasive grains, the watershed segmentation algorithm is useful for separating larger objects into smaller ones. This method can extract grains quite well, but may not distinguish as separate two close vertices of similar height belonging to different grains. It is then possible to apply the vertex extraction procedures first and then use the results of the watershed segmentation algorithm [21].

Analysing the chip and side pile-ups formation mechanism, it can be concluded, that grain orientation with respect to the direction of grinding and increasing the width of the cut layer is one way to reduce unfavourable lateral material flows. Reduction of lateral material flows occurs for grains having a flat or more favourably concave contact surface with a small rake angle γ (usually negative in abrasive micro-cutting processes). Vertex features such as large radii of rounding are unfavourable, and even less favourable are vertex abrasion and loading of the tool surface with machining products.

Vertex parameters are particularly important in the assessment of surfaces. They are active areas of the grinding wheel in contact with the workpiece. Therefore, from the point of assessment of the cutting ability of the grinding wheel, special attention should be paid to the features of vertices located in a certain range of ordinates, e.g., from 0.1 to 0.4 of the value of the St parameter from the highest surface vertex (Figure 12). 

Valuable conclusions regarding the results of the abrasive machining process can be drawn, among others, from the division of the tool surface using Voronoi cells. Each Voronoi cell contains one surface vertex (abrasive grain). The division of the surface carried out in this way makes it possible to determine:the distance between the vertices using the nearest neighbour method [36];the relationship between the area of the abrasive grain base and the area of the Voronoi cell;the inclination angles of the lines connecting adjacent grain vertices.

Evaluation of the distribution and shape of grain vertices’ elevations above a certain level from the highest apex can be the basis for, among other things:assessing grain activity,predicting the specific energy of machining, which depends on the distribution of geometric parameters of layers cut by individual grains,evaluation of tool wear processes,selection of machining parameters.

Important control features include:area of elevation over a specific plane,area of the base of the elevation, geometric centres of the elevations, number of elevations,the height and height dispersion of the tops of the elevations,the average distance between the vertices of the hills, determined from the decomposition of the surface into Voronoi cells and the area of the Voronoi cells.

The geometric parameters of the characteristic features of the active surface of the grinding wheel can be used to identify and classify the forms of wear. In the classification process, it must be taken into account that the abrasive wear area of the abrasive grains is much smaller than the cross-sectional area of the abrasive grain. Flat areas, resulting from the clogging of the grinding wheel surface, are characterized by a larger area and circumferential development. In the example described below for the identification of abrasive wear and loading on the surface of grinding wheels, the following grinding wheels manufactured at ANDRE ABRASIVE ARTICLES (Koło, Poland) were used (Table 1, Figure 13). The parameters of the grinding process are given in Table 2.

A sliding analysis window was used by performing detection of flat areas (Figure 14) on the active surface of the grinding wheels. The histograms (Figure 15) indicate the range of occurrence of flat areas with an area larger than the largest cross-sectional area of the abrasive grain. These are instances of loading on the surface of the grinding wheel. In addition, the coefficient of elongation of flat areas (large deviation from circularity) on the active surface of the tested grinding wheels was analysed, as a second parameter for differentiating abrasive grain vertices abrasion from abrasive grain pulling-out on the grinding wheel surface. It was shown that B-type grinding wheels with a modified spatial structure showed the least tendency to be affected by active surface loading during the grinding process. The highest percentage of loading on the grinding wheel surface after the grinding process was observed for a tool containing special grains with a spatially developed structure. The identification of planar objects allows additional analysis of the wear state of the abrasive tool. It was pointed out that the commonly used parameters for assessing the height of surface irregularities do not sufficiently differentiate the degree of wear of abrasive tools.

## 3. Dedicated Parameters for Evaluating the Geometric Structure of the Grinding Wheel Surface

The previously described measures for assessing the condition of the active surface of abrasive tools contain characteristics that are important, but their information content is only magnified in a properly selected set of several measures. In the case of the evaluation of the active surface of abrasive tools, the evaluation of the features that have the greatest impact on the results of the process and the prediction of tool life is of particular importance. These features relate primarily to the shape and position of abrasive grain vertices. The parameters proposed in the paper [37] for evaluating the relevant features of the active surfaces of abrasive tools were developed using lessons learned from applications in signal analysis of the Teager–Kaiser energy operator (TKEO) Ψz as a useful signal demodulation [38]. This operator was first proposed by Teager and subsequently refined by Kaiser. It captures the level and variability of the signal being analysed. In describing the above operator, Kaiser cites the analogy of the energy of an oscillating spring-mass system, in which the total energy is proportional to the square of the velocity of the mass and its potential. This operator is defined as follows:Ψz(z(x)) = (dz/dx)^2^ − z(x) × (dz^2^/dx^2^)(1)

The discrete value of the TKEO operator can be estimated from three adjacent samples of the signal z_i−1_, z_i_, z_i+1_, (for all values z of the signal z, i.e., for i = 1…n).
Ψz(z(i)) = z(i)^2^ − z(i − 1) z(i + 1)(2)

In the assessment of the active surface of abrasive tools, the geometric parameters of the areas covering the tops of grains are of great importance. This assessment is made using the cut-off surface at a specific level h. In the case of evaluating the height and variability of hill heights, it is reasonable to determine the absolute value of Ψz. The usefulness of the Ψz index for surface assessment, even after these changes, is limited. This is due to the determination of the value of this parameter from the surface profile. This reduces the informative usefulness of the Ψz indicator in the analysis of the topography of the grinding wheel surface.

The authors of [37] therefore developed an index of the height and variation of the height (sharpness) of grain vertices related to the surface (3D), Shs, described in detail in the publication. It is also possible to recommend an analysis using a parameter which is the product of the geometric mean of the gradient in the x and y directions, and the height of a given point, determined with a step equal to the average width of the vertices, representing elevations above the level distant by the depth of grinding from the topmost vertex in the studied section of the grinding wheel surface.

## 4. Methodology for Testing and Evaluation of Abrasive Grains Wear Processes and Changes in Tool Shape and Dimensions

In the study of grinding wheel wear processes caused by the phenomena of abrasive and fracture wear of abrasive grains, it should be taken into account that the grinding and wear processes of the grinding wheel depend on the variable state of the active surface of the wheel and are not stationary processes. On the basis of the conducted studies of grinding wheel life, the following conclusions can be made about the diagnostics of the intensity of shaped and volumetric wear of abrasive tools:the processes of chipping caused by exceeding the strength of the grain and chipping caused by exceeding the resistance to total load occur in parallel;grains with longer operating time are characterised by greater abrasion, which causes an increase in load;the process of spalling due to exceeding the strength depends on the distribution of the grain’s limiting strength and the distribution of grain load, varying during successive contacts of the grain with the workpiece;the intensity of chipping of grains in the first stage of the grinding process decreases with the passage of grinding wheel operation, especially when coarse shaping is used;the variable intensity of strength wear depends on the characteristics of the abrasive tool, in particular, on the abrasive material and bond used, as well as their share in the volume of the grinding wheel;in the process of grinding with grinding wheels of low hardness, the process of chipping of whole abrasive grains prevails, while in the case of higher hardness of the tool, the processes of microchipping of grain fragments or increasing abrasive wear of grain vertices become more important;chipping of active grains causes a reduction in the total area of abrasion of all active grains;the number of grains chipped at a given moment is a time-dependent value;under operating conditions with intensive self-sharpening, the probability of grain chipping does not depend on the operating time of the grain;under operating conditions of a grinding wheel with limited self-sharpening, the average waiting time for chipping in a group of grains with a fixed working time, may depend on the total working time of the abrasive grains;the distribution of working time of active grains varies with the grinding time;constant chipping intensity can be described by an exponential distribution of grain life;decreasing chipping intensity can be described by a logarithm-normal distribution of grain life;increasing chipping intensity can be described by a normal one-sided truncated grain life distribution or a Weibull and gamma distribution.

## 5. Prediction Wheel Life and Productive Grinding Operations

Determining the life of a grinding wheel requires an assessment of whether a specific set of conditions describing acceptable changes in the characteristics of the process or results of the machining operation are met. These changes are mainly due to the wear processes of the active grains and the resulting changes in the properties of the tool. The output quantities of the grinding process *y_i_* are functions of the grinding parameters *x_i_*_1_, measurable but uncontrollable quantities *w_i2_* and disturbances *z_i_*_?_, the number and impact of which are unknown.
(3)$$y_{j}=f_{i}\;[w_{i}{}_{2}(t),\;\;z_{i}{}_{?}]\;|{}_{x_{i}{}_{1}=const}$$

For typical grinding processes, the set of limiting conditions does not include the values of the process parameters, because these are invariant. The conditions refer to the normalised Kproc process characteristics and the results of the Koper machining operation, defined so that decreasing their values mean decreasing the period that remains until the end of the life:(4)$$K_{s}=\min(\min(K_{proc}),\min(K_{{}_{oper}}))$$
where *K_s_* the value of the feature whose level is closest to the limit of the permissible value.

In doing so, it is assumed that for certain grinding parameters *x_i_*, the assumed lifetime *T_s_(x_i_)* will be such that the limiting conditions imposed on the selected quantities *y_j_* will be satisfied throughout this time. In the period from the beginning of machining to the tool life, the progressive wear of the active surface of the tool causes changes in at least one of the conditions *w_ik_(t)*.
(5)$$T_{j}\leq T_{y_{j}}(y_{jk\;\;dop})\;,x_{i1}=const$$

The lifetime of the grinding wheel *T_s_*, in which all the recorded and normalised process features and machining results are within the limits of their acceptable deterioration, is determined by the relation:(6)$$T_{s}=\min[T_{K_{s}}\;(K_{s\;\;\min}),\;\;T_{y_{j}}\;(y_{j\;\;dop})]$$

This means that the life of the grinding wheel *T_s_* is equal to the sum of the times of grinding operations after which at least one of the process characteristics or characteristics describing the results of the process reaches the limiting value (Figure 16). The magnitudes *y_j_*, on which the limiting conditions are imposed, depend on the grinding parameters *x_i_*, the grinding wheel operation time *t* since the last shaping of its active surface and interferences *z*, the number and impacts of which are unknown. From the limiting conditions:(7)$$y_{j}=f_{o}\left(x_{i},t,z\right)\geq \{\left(\leq \right)\}y_{idop}$$

It follows that the times *tj* after which the quantities *y_j_* have already reached the limit values will depend on *x_i_*, *y_j dop_* and interference *z*. This means that in successive realisations, the times *t_j_*_min_
*=* min*(t_j_)* will be different. The functions *T_Ks_* and *T_yj_* contain random components that depend on the factors described earlier, mainly on the variation of the machining allowance in successive operations, the properties of the material being machined and the feeding characteristics and properties of the machining fluid.

From the dependence of the normalised process features *c*1, *c*2 and *c*3 as a function of the feature grinding time, it can be seen that under conditions of anticipatory renewal of the tool surface, the life of the grinding wheel is most often determined by changes in the feature *c*1 and then the life is less than 40 min. The life is then determined on the basis of the predetermined probability of exceeding the permissible value, without checking whether in a given realisation of the process the permissible value is exceeded by the evaluated features. In the case of supervised process condition, the life of the grinding wheel is most often determined by changes in the *c*2 feature, and then the average tool life exceeds 60 min.

A diagram for analysing the problem of determining durability is shown in Figure 17.

During supervision, it is checked whether the values of the monitored features are exceeded, and then statistically the life corresponds to the average value from all realisations. Durability distributions that were determined taking into account each characteristic separately will, of course, differ in expected value and variance. Most often, it can be assumed that the form of the distribution, as a result of the multiplicative mechanism of cumulative influence of factors affecting changes in individual features, will be close to a log-normal distribution. From the above observations, it can be seen that under machining conditions in which the condition of the tool and the process features are monitored, in successive periods of use of the grinding wheel, the service life may be determined by various limiting conditions, but the average service life will be at a level significantly higher than that corresponding to the anticipatory renewal of the active surface of the tool.

In the diagnosis of grinding processes, assessments of the production efficiency of grinding operations *W* and its cost *K* cannot be ignored. These quantities depend on the life of the grinding wheel and the machining parameters. For productivity *W*, the relationship is as follows:(8)$$W=\{\frac{t_{pz}}{n}+\left[t_{p}+t_{g}\left(Q_{v}\right)\times \left(1+\frac{t_{k}}{T_{s}\left(Q_{v},W_{g}\right)}\right)\right]\times \left(1+k_{u}\right){\}}^{-1}$$
where *t_pz_* represents the preparation and completion time, *n* is the number of items in the series, *t_p_* is the auxiliary time, *t_g_*(*Q_v_*) is the main time depending on the volumetric efficiency of grinding *Q_v_*, *t_k_* is the time of shaping the surface of the grinding wheel, *k_u_* denotes the normative of supplementary time and *T_s_*(*Q_v_,W_g_*) denotes the life of the grinding wheel depending on *Q_v_* and a set of limiting conditions *W_g_*, imposed on the machining operation.

The dependence of the main time *t_g_* on the volumetric machining capacity *Q_v_* is a deterministic dependence, while the dependence of the grinding wheel life *T_s_* on both the capacity *Q_v_* and the limiting conditions *W_g_* imposed on the process and machining results is a stochastic dependence. In a typical situation where several process features or machining results are used to evaluate the life of a grinding wheel under supervised conditions, it must be taken into account that in subsequent life periods, the time remaining to renew the tool surface may be determined by features different from those of the previous period. This contributes to the considerable complexity of the decision-making process in these tasks. Supervising the condition of a tool significantly improves the utilisation of the tool’s machining potential during each tool life, which means lower tooling costs, lower machining costs and increased productivity as a result of reduced tool surface renewals.

## 6. Directions of Abrasive Tool Development

The observed development of abrasive machining processes has involved the development of new abrasive materials and bonding agents, the improvement of abrasive tool features, the introduction of new machining methods, more efficient and economical methods of using machining fluids, new machine tools and the spread of supervisory methods. As a result, significant results are being achieved in improving surface quality and machining accuracy and efficiency. New machining methods are being developed and the range of applications for high-precision machining is being greatly expanded. New tools increasingly have a functionally locally differentiated structure.

It can be predicted that the near future will see the development of abrasive tools with a multiphase structure, i.e., hybrid tools, in which the conventional porous structure consisting of grains and binder will be modified by additions of other phases in the form of abrasive microaggregates, local bands or layers. Applications of 3D printing to produce special abrasive tools can also be expected. For example, by introducing new abrasive microaggregates into the structure of bonded abrasive tools, there is a quantum leap in the adaptability of tools to the requirements of various technological operations. The tools currently in use can be differentiated by changes within certain limits concerning only the type and size of abrasive grains, bond and structure. In the new tools, it will be possible to further differentiate tool features by selecting the size of microaggregates, grains and binders that make up these aggregates, as well as by varying the types of aggregates and their proportion in the grinding wheel. The use of micro-aggregates makes it possible to achieve adaptive grinding wheel structures, as a different mechanism of micro-aggregate self-sharpening can be used to spontaneously create increased porosity on the active surface of the tool in the form of pits. It is also possible to locally (zonally) vary the proportion and type of aggregates in the tool volume.

Along with microaggregates, additives may be introduced to reduce the specific energy of grinding, additives to reduce the effects of the environment on the machined surface, and additives conducive to obtaining special structures in the form of regular macrogeometry, with specific operational or decorative significance. In subsequent phases of development, microaggregates may also, in applications for particularly important operations, contain substances used for process diagnostic purposes and tool condition evaluation, inducing, for example, a change in the colour of the machining fluid or the workpiece surface. The biggest development challenge in the near term may be the manufacture of abrasive tools that would have built-in sensors for evaluating grinding power or temperature, as well as wireless transmitters of control signals.

## 7. Conclusions

Comprehensive diagnostics of grinding processes includes many features, the analysis of which requires consideration of the probabilistic basis of inaccuracies in the shaping of workpieces, random variation, fluctuations and complex mechanisms of phenomena in abrasive tool wear processes.

The methodology for the study and evaluation of the condition of the active surface of tools is one of the complex issues, as it requires evaluations derived from complementary sets of parameters, with typical parameters for evaluating the geometric structure of the surface in this application not having sufficient capacity to differentiate the condition of the tool surface. Acquisition of data describing the surface without removing the grinding wheel from the spindle through optical inspection systems or surface replica techniques is being sought.

This publication identifies new dedicated parameters for evaluating grinding wheel surface properties with a high ability to differentiate changes occurring as a result of abrasion of grain vertices, their chipping or loading of the grinding wheel surface. Such parameters are, among others, an index that integrates the influence of height and sharpness of abrasive grain vertices. The described parameters are characterised by higher informational usefulness, meet the condition of complementarity, and facilitate the determination of possible corrections to the process of shaping the surface of grinding wheels and the prediction of their service life.

In the diagnosis of grinding processes, analyses of the specific energy of machining and ways to reduce the energy intensity of the processes play an important role. The purpose of diagnostic procedures is to ensure the required quality and efficiency of machining. This publication also presents abrasive tool development directions concerning hybrid tools with a multi-phase structure, in which the conventional porous structure consisting of grains and bonds will be modified by additions of other phases in the form of abrasive microaggregates, local bands or layers. The use of incremental techniques (3D printing) to produce specialised abrasive tools, including those with built-in process sensors, can also be envisioned.

## Figures and Tables

**Figure 1 materials-16-01493-f001:**
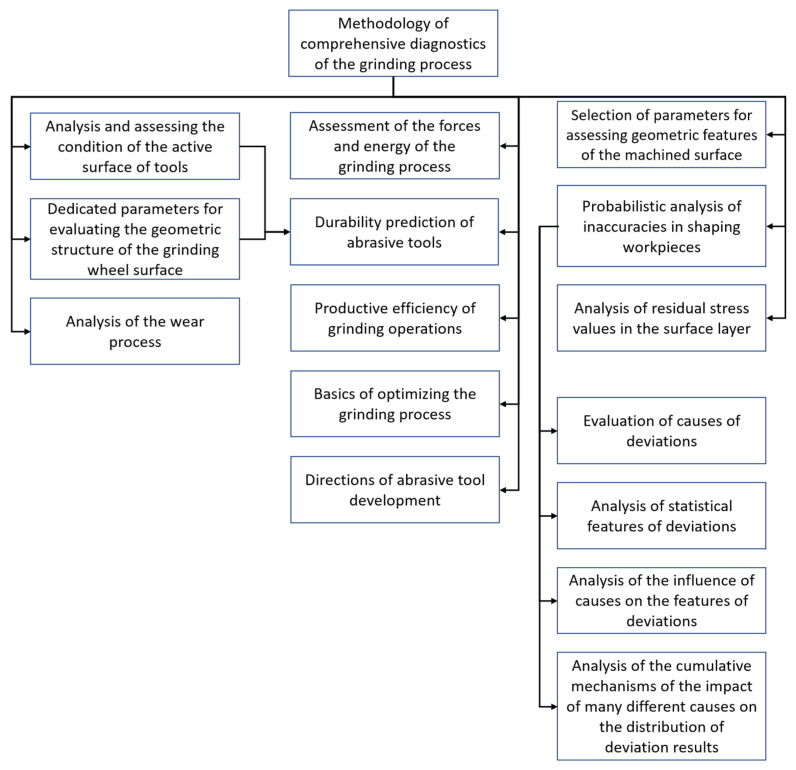
Diagram for methodology and comprehensive diagnostics of the grinding process.

**Figure 2 materials-16-01493-f002:**
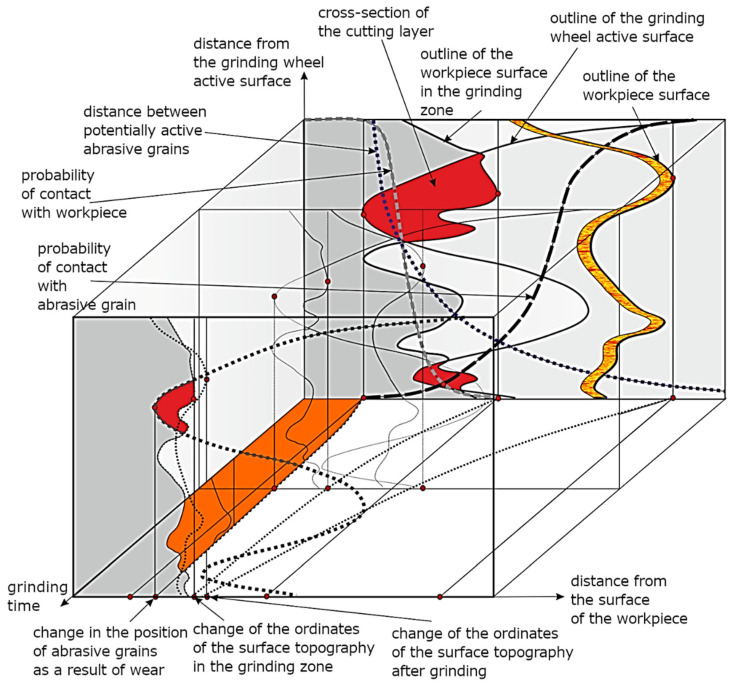
Diagram for analysing possible contacts of grains with the workpiece in the grinding process.

**Figure 3 materials-16-01493-f003:**
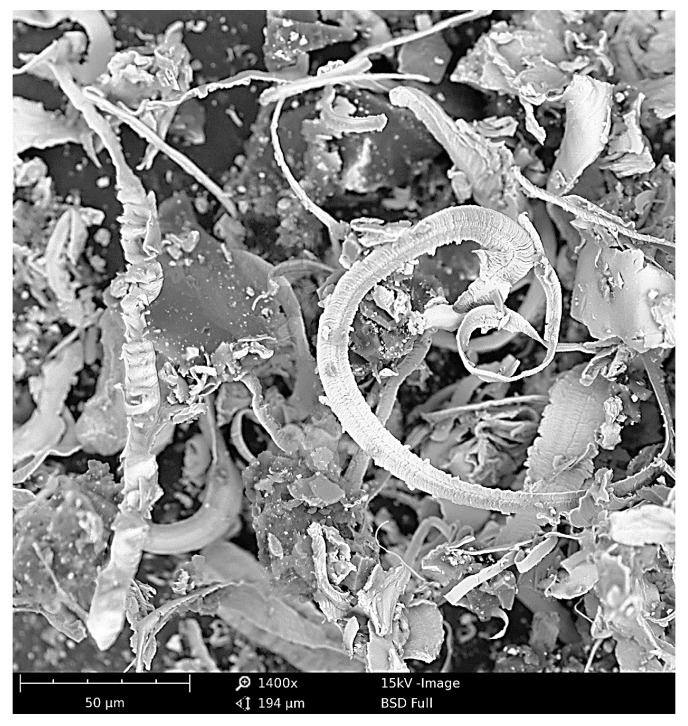
SEM images of microchips formed in the grinding process of X153CrMoV12-bearing steel (NC11LV).

**Figure 4 materials-16-01493-f004:**
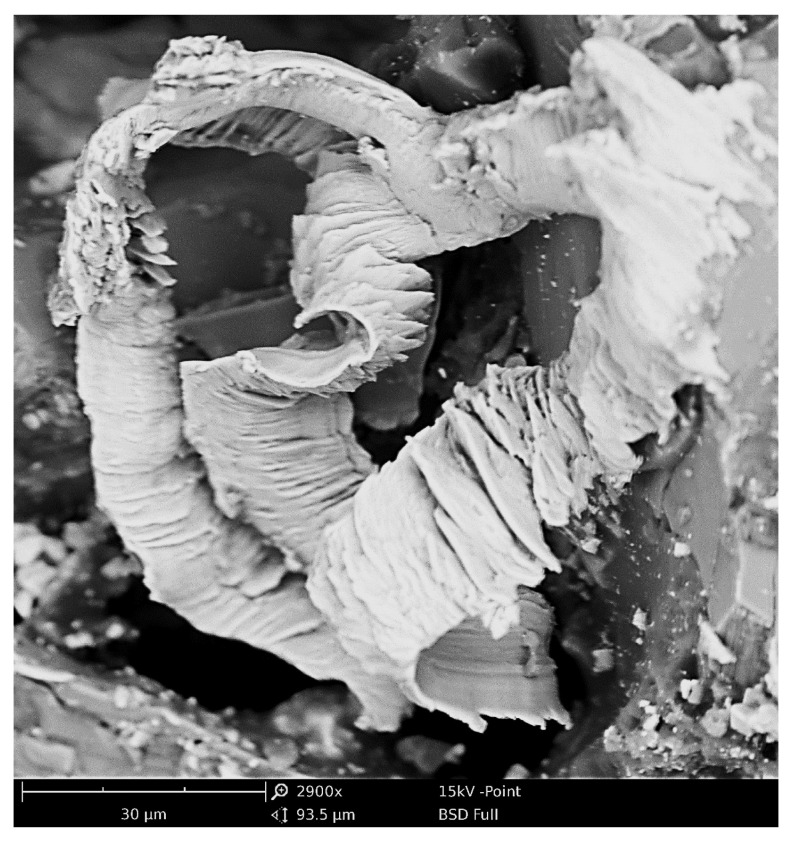
SEM images of loading and microchips on the surface of the grinding wheel in the grinding process of Grade 5 titanium alloy.

**Figure 5 materials-16-01493-f005:**
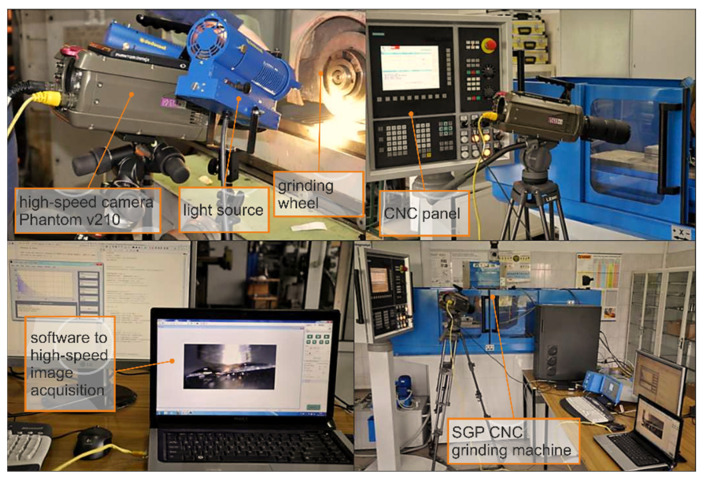
Workstation for acquisition of high-speed images during the micro-cutting process.

**Figure 6 materials-16-01493-f006:**
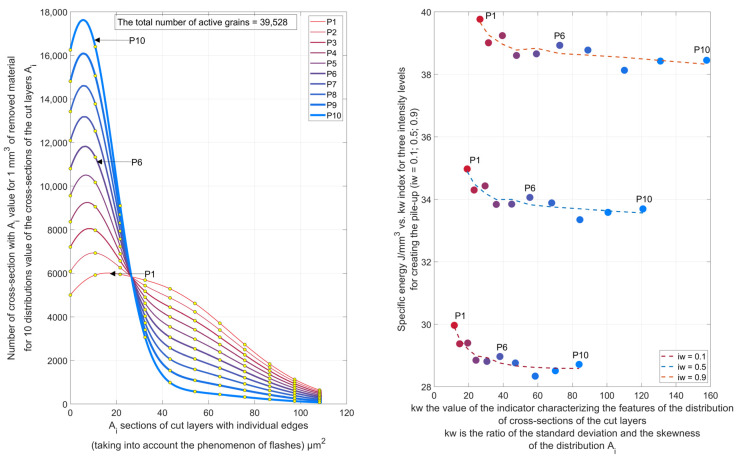
Illustration of the effect of the form of the distribution of the cross-sections of the layers cut by the individual edges on the unit energy of grinding for three cases with different intensities of efflorescence formation.

**Figure 7 materials-16-01493-f007:**
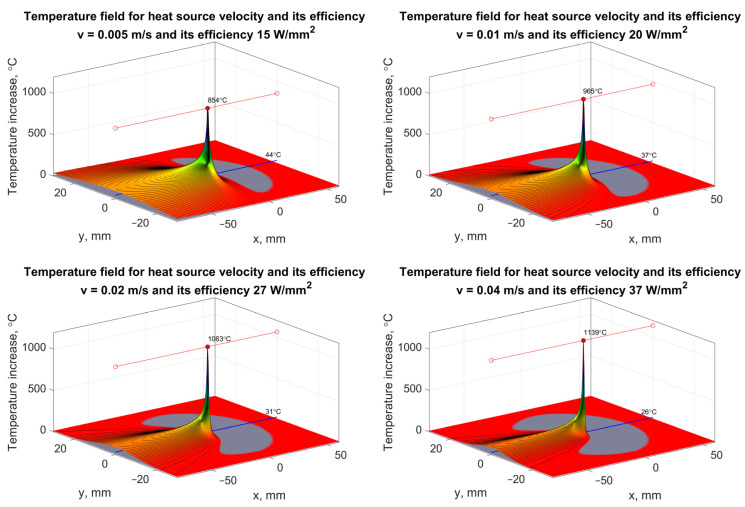
Temperature field during grinding of flat components made of construction steel for different speeds of heat source and speed-dependent heat source efficiency.

**Figure 8 materials-16-01493-f008:**
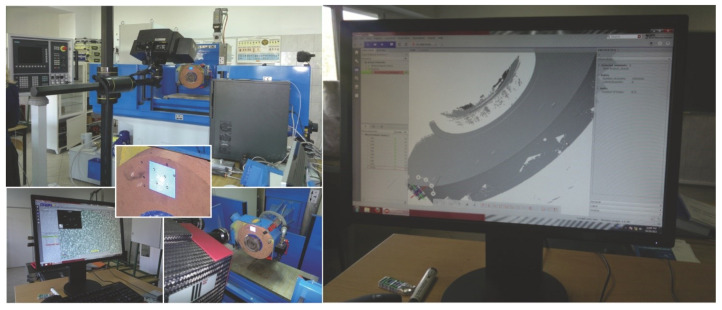
Stand for scanning the surface of the grinding wheel remaining on the grinding machine spindle.

**Figure 9 materials-16-01493-f009:**
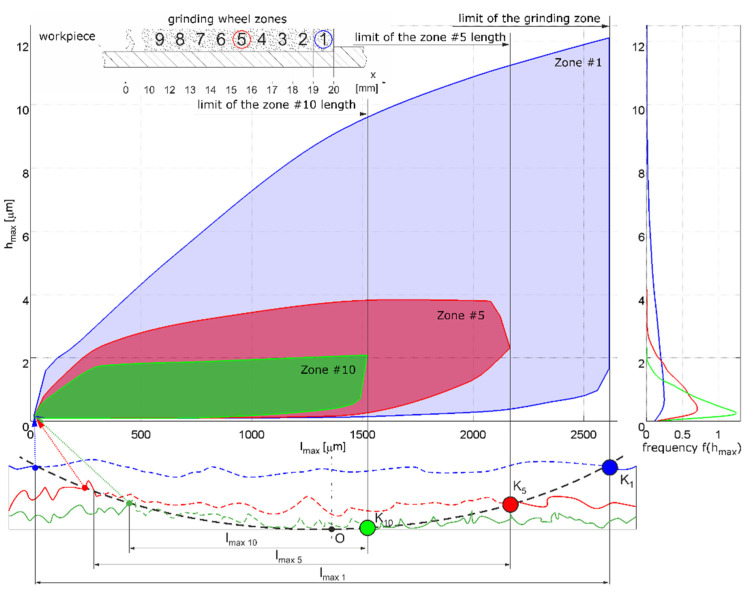
Grinding zone boundaries for isolated ridges on the active surface of the grinding wheel (numbers of ridges are defined from the active edge).

**Figure 10 materials-16-01493-f010:**
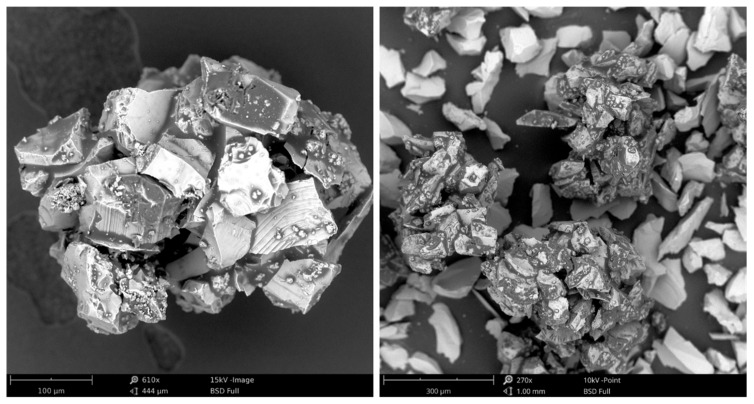
Microscopic images of microaggregates of grains and separate grains in specific proportions used in the developed tools with abrasive microaggregates.

**Figure 11 materials-16-01493-f011:**
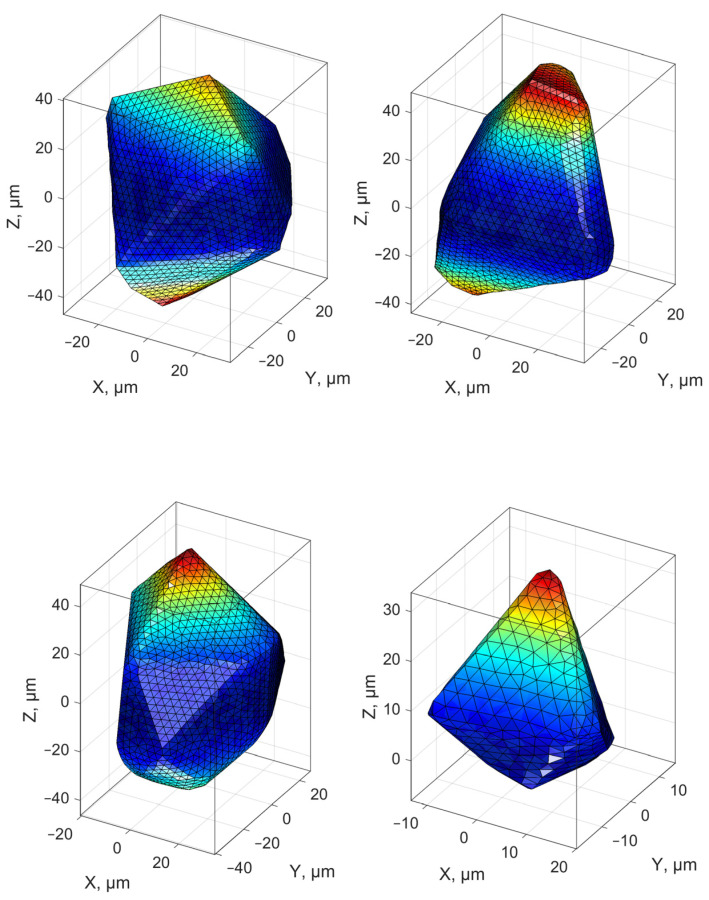
Visualisations of abrasive grain geometry as an illustration of the diversity of shape and position of the vertices (top image using triangulation, bottom images with contours plotted).

**Figure 12 materials-16-01493-f012:**
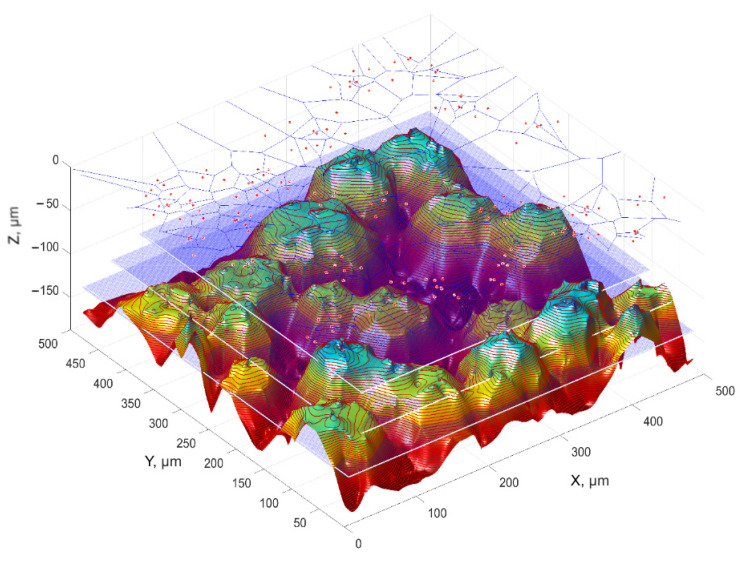
A scheme for analysing of the elevation characteristics of the grain vertices located at different levels of *St* values from the highest ordinate of the grinding wheel.

**Figure 13 materials-16-01493-f013:**
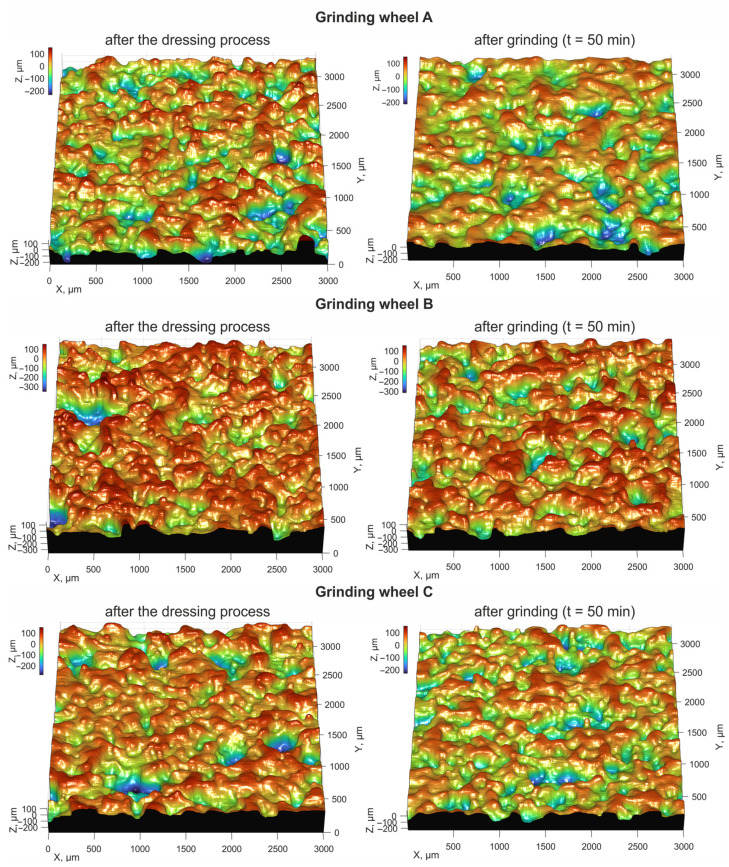
Images of the surface topography of the tested grinding wheels after the dressing process and after 50 min of grinding.

**Figure 14 materials-16-01493-f014:**
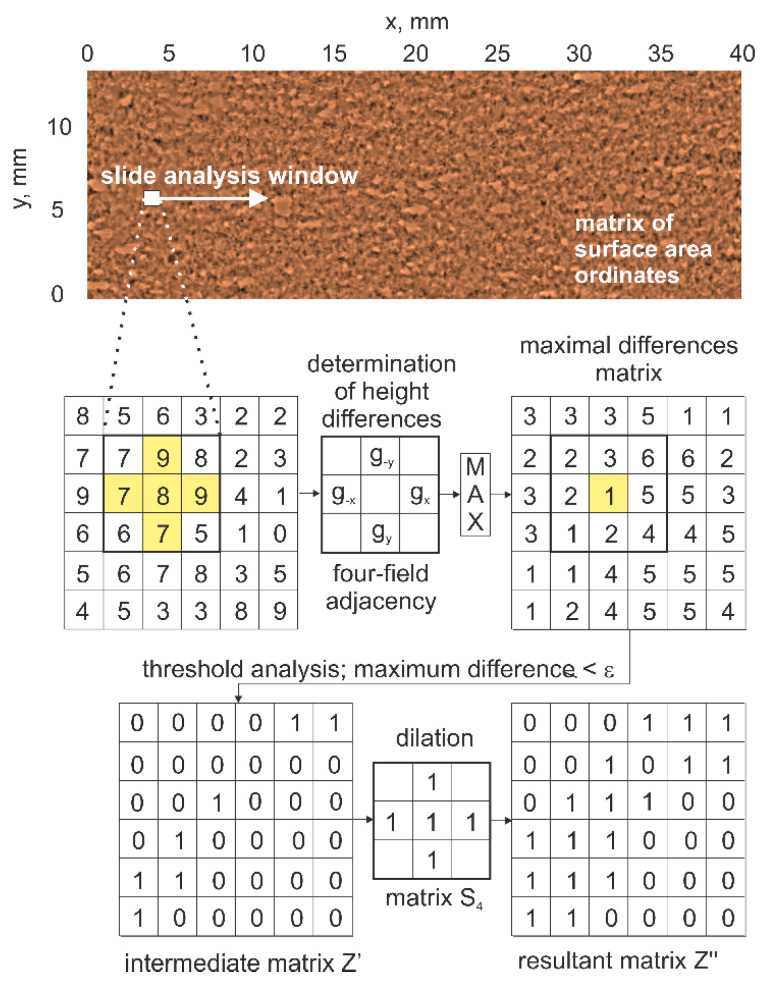
Flat area detection scheme for four-field neighbourhood.

**Figure 15 materials-16-01493-f015:**
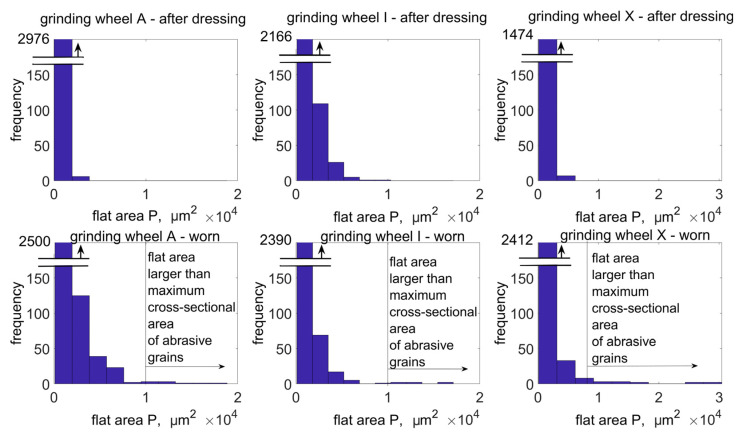
Histograms of Pi areas of flat areas identified on the active surface of the tested grinding wheels.

**Figure 16 materials-16-01493-f016:**
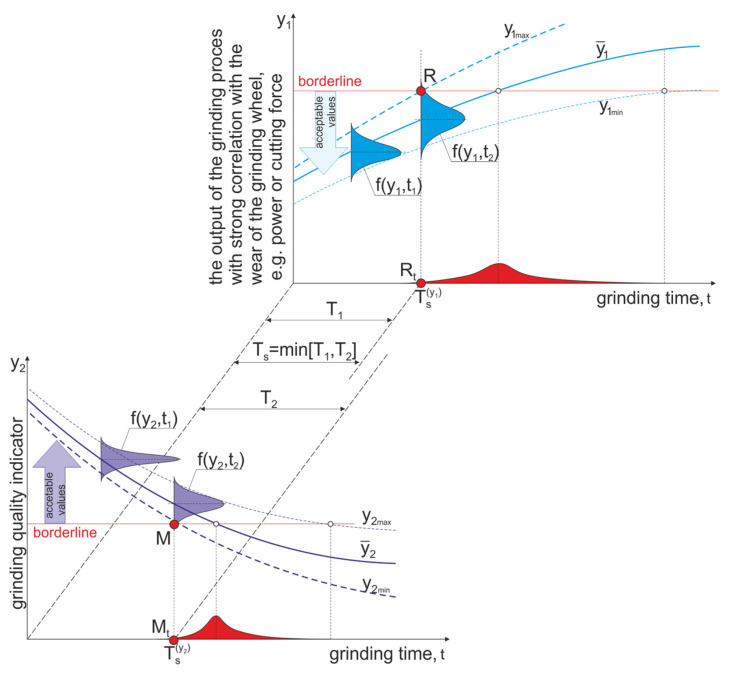
Diagram to illustrate the problem of determining tool life taking into account probabilistic features of the grinding wheel wear process.

**Figure 17 materials-16-01493-f017:**
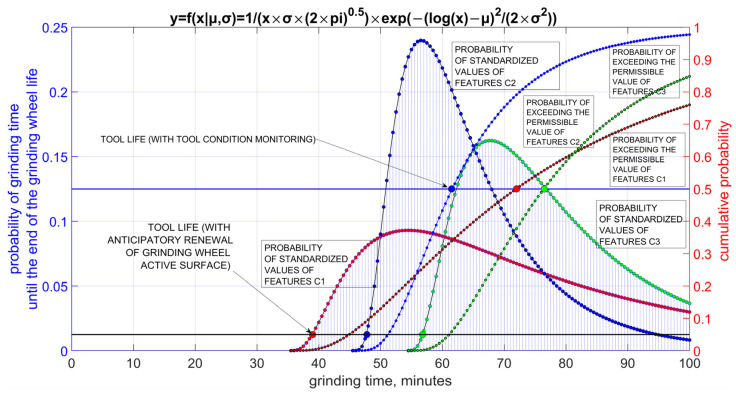
Distributions of values of features considered in the evaluation of grinding time to end of life on the basis of control of various process features for two strategies: anticipatory renewal of the tool surface and supervision of the process condition, where *c*1—normalised value of grinding force, *c*2—normalised value of parameter *Sp*5 of the height of surface roughness, *c*3—normalised value of the deviation of the diameter of the ground cylindrical surface.

**Table 1 materials-16-01493-t001:** Characteristics of grinding wheels for comparison tests.

A	1–250 × 25 × 76, 2-99A100K7VTE10-35 Conventional grinding wheel
B	1–250 × 25 × 76, 2-99A100K7IVTE10-35 Grinding wheel with modified spatial structure (with increased porosity)
C	1–250 × 25 × 76, 2-99AY100K7VTE10-35 Grinding wheel containing special grain with spatially developed structure

**Table 2 materials-16-01493-t002:** Parameters and conditions of grinding process.

	Parameter	Unit	Value
1	Peripheral speed of the grinding wheel	m/s	35
2	Dressing feed	mm/rev.	0.1–0.15
3	Longitudinal feed	m/min	25
4	Transverse feed	mm/stroke	0.5
5	Workpiece material	-	X153CrMoV12 (NC11LV)

## Data Availability

The data presented in this study are available from the corresponding author, M.W.P., upon reasonable request.

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
