# Peer review of "Technological Aspects of Variation in Process Characteristics and Tool Condition in Grinding Process Diagnostics"

_materials, 2023, doi:10.3390/ma16041493_

Round 1
Reviewer 1 Report
1. The study contain many contents. In order to improve its readability, it is recommended to supplement “introduce section” to explain the background and the goal of the manuscript. And a flowchart is suggested to be provided to illustrate the logical relationship of every part.
2. The abstract is expected to be more concise. For example, some sentences from line 11 to 18 are simple statement for grinding phenomenon, which can be given in introduce section, not in abstract.
3. The figure 5 is not clear, lacking necessary illustrations.
4. If the formulas and figures are cited from other’s research, corresponding references should be cited and marked.
5. The references should be marked sequentially.
6. The conclusion must be concise and improved.
Author Response
Dear Reviewer, thank you for considering our manuscript for publication in Materials. We are grateful for the valuable suggestions provided. In the revised manuscript, all changes are highlighted in red.
Below, we submit responses to the comments.
Point 1: The study contain many contents. In order to improve its readability, it is recommended to supplement “introduce section” to explain the background and the goal of the manuscript. And a flowchart is suggested to be provided to illustrate the logical relationship of every part.
Response 1: At the beginning (first section), information about the scope and issues raised in the work was added. A diagram has been included that illustrates the substantive link between the subsections of the article.
Point 2: The abstract is expected to be more concise. For example, some sentences from line 11 to 18 are simple statement for grinding phenomenon, which can be given in introduce section, not in abstract.
Response 2: The abstract has been revised and shortened in accordance with the comments.
Point 3: The figure 5 is not clear, lacking necessary illustrations.
Response 3: Descriptions have been added to the figure, indicating the connections between the data in the left and right charts. The colors of the points on the right graph have been corrected and a legend has been added for the values ​​indicated as the intensity for creating side ridges (iw = 0.1; 0.5; 0.9).
Point 4: If the formulas and figures are cited from other’s research, corresponding references should be cited and marked.
Response 4: Corrections have been made to the article in accordance with the comments.
Point 5: The references should be marked sequentially.
Response 5: The order of references has been corrected
Point 6: The conclusion must be concise and improved.
Response 6: The summary has been corrected.
Reviewer 2 Report
1. The abstract is too long and needs to be condensed.
2. The keywords should be added.
3.If the Fig.1,2,3 is cited, please identify the relative literature.
4. The logic of this article seems unreasonable.
5.The conclusions are to be revised and rewritten for their better reflection.
Author Response
Dear Reviewer, thank you for considering our manuscript for publication in Materials. We are grateful for the valuable suggestions provided. In the revised manuscript, all changes are highlighted in red.
Below, we submit responses to the comments to the review.
Point 1: The abstract is too long and needs to be condensed.
Response 1: The abstract has been revised and shortened in accordance with the comments.
Point 2: The keywords should be added.
Response 2: The keywords has been added
.
Point 3: If the Fig.1,2,3 is cited, please identify the relative literature.
Response 3: Corrections have been made to the article in accordance with the comments.
Point 4: The logic of this article seems unreasonable.
Response 4: The abstract has been modified to highlight the main points of the article.
At the beginning (first section), information about the scope and issues described in the article was added. A diagram has been included that illustrates the substantive link between the subsections of the article
Point 5: The conclusions are to be revised and rewritten for their better reflection.
Response 5: The conclusion has been corrected.
Reviewer 3 Report
Authors considered the parameters used to evaluate the topography of ground surfaces do not have sufficient capabilities to differentiate the surface condition of grinding wheels. Also, a new research direction has been formulated on the use of 3D printing to produce specialized abrasive tools, including those with built-in process sensors. The article is interesting
1. The Findings needs to be more validated with existing literature.
2. The conclusions need to be reformulated.
3. Line 42 may be written as Introduction as heading.
4. Needs to re look at introduction section. Citations are not in proper order.
5. As per the Fig.2 and 3 microdiscontinuities in chip formation has been formed. Provide the reason.
6. Provide the parts in Figure 4.
7. On what basis Peripheral speed of the grinding wheel, Forming feed, Longitudinal feed, Transverse feed has been selected in Table 2.
8. Authors should strictly follow the format of the journal.
Author Response
Dear Reviewer, thank you for considering our manuscript for publication in Materials. We are grateful for the valuable suggestions provided. In the revised manuscript, all changes are highlighted in red.
Below, we submit responses to the comments to the review.
Authors considered the parameters used to evaluate the topography of ground surfaces do not have sufficient capabilities to differentiate the surface condition of grinding wheels. Also, a new research direction has been formulated on the use of 3D printing to produce specialized abrasive tools, including those with built-in process sensors. The article is interesting.
Thank you for your positive opinion about our article.
Point 1: The Findings needs to be more validated with existing literature.
Response 1: The article has been corrected in accordance with the comments. References to findings have been added and tidied up
Point 2: The conclusions need to be reformulated.
Response 2: The conclusion has been corrected.
.
Point 3: Line 42 may be written as Introduction as heading.
Response 3: Added changes to the heading in section 1 as noted
Point 4: Needs to re look at introduction section. Citations are not in proper order.
Response 4: At the introduction, information about the scope and issues described in the article was added. A diagram has been included that illustrates the substantive link between the subsections of the article. The order of references has been corrected
Point 5: As per the Fig.2 and 3 microdiscontinuities in chip formation has been formed. Provide the reason.
Response 5: The lamellar, segmental structure of microchips formed during the microcutting process results from the high frequency of cyclic stress variation. The thickness of the chip plates depends on the type of material and the energy absorbed by plastic deformation. The greater the share of plastic deformation in the cutting process, the smaller the thickness of the chip plates, and the greater the frequency of their formation.
Point 6: Provide the parts in Figure 4.
Response 6: Figure 4 has been changed as noted.
Point 7: On what basis Peripheral speed of the grinding wheel, Forming feed, Longitudinal feed, Transverse feed has been selected in Table 2.
Response 7: Typical parameters for machining this type of alloy were used. Added a reference to the table in the article.
Point 8: Authors should strictly follow the format of the journal.
Response 8: The formatting of the article has been corrected in accordance with the journal's recommendations.